# Graphene Oxide Surface Modification of Reverse Osmosis (RO) Membrane via Langmuir–Blodgett Technique: Balancing Performance and Antifouling Properties

**DOI:** 10.3390/membranes14080172

**Published:** 2024-08-07

**Authors:** Dmitrii I. Petukhov, James Weston, Rishat G. Valeev, Daniel J. Johnson

**Affiliations:** 1Division of Engineering, New York University Abu Dhabi, Abu Dhabi P.O. Box 129188, United Arab Emirates; dp3916@nyu.edu; 2Water Research Center, New York University Abu Dhabi, Abu Dhabi P.O. Box 129188, United Arab Emirates; 3Core Technology Platforms, New York University Abu Dhabi, Abu Dhabi P.O. Box 129188, United Arab Emirates; james.weston@nyu.edu; 4Udmurt Federal Research Center of the Ural Brunch of Russian Academy of Sciences (UdmFRC of UB RAS), Izhevsk 426067, Russia; rishatvaleev@udman.ru

**Keywords:** graphene oxide, Langmuir–Blodgett, surface functionalization, fouling, reverse osmosis

## Abstract

The reverse osmosis water treatment process is prone to fouling issues, prompting the exploration of various membrane modification techniques to address this challenge. The primary objective of this study was to develop a precise method for modifying the surface of reverse osmosis membranes to enhance their antifouling properties. The Langmuir–Blodgett technique was employed to transfer aminated graphene oxide films assembled at the air–liquid interface, under specific surface pressure conditions, to the polyamide surface with pre-activated carboxylic groups. The microstructure and distribution of graphene oxide along the modified membrane were characterized using SEM, AFM, and Raman mapping techniques. Modification carried out at the optimal surface pressure value improved the membrane hydrophilicity and reduced the surface roughness, thereby enhancing the antifouling properties against colloidal fouling. The flux recovery ratio after modification increased from 65% to 87%, maintaining high permeability. The modified membranes exhibited superior performance compared to the unmodified membranes during long-term fouling tests. This membrane modification technique can be easily scaled using the roll-to-roll approach and requires minimal consumption of the modifier used.

## 1. Introduction

Membrane technology has carved out a unique niche in addressing the global water scarcity problem. For instance, over 65% of freshwater worldwide is generated through reverse osmosis (RO) treatment of sea or brackish water [1]. The widespread adoption of reverse osmosis is attributed to its significantly lower energy consumption than distillation techniques. However, fouling is a common challenge across membrane processes, including reverse osmosis. Fouling occurs when insoluble salts, macromolecules, colloids, organic molecules or particles deposit on the membrane’s surface or within its pores, diminishing performance, increasing energy consumption, and raising treatment costs. Various strategies can be employed for fouling mitigation, such as feed stream pretreatment, optimization of process parameters, or membrane modification. Modification, in particular, allows alteration of the membrane surface’s physicochemical properties, such as the roughness, hydrophilicity, or surface charge, significantly impacting the foulant deposition rates [2]. Hydrophilic polymers [3,4], zwitterionic materials [5,6], inorganic materials [7], and carbon-based [8,9] nanomaterials have been utilized for the modification of RO membranes. During the last decade, carbon nanomaterials and, especially, graphene oxide (GO) have been considered promising substances for membrane modification due to their unique properties, such as their high hydrophilicity, chemical stability, ease of synthesis and low cost of production at less than USD 5 per gram [10].

The antifouling properties of RO membranes can be enhanced by incorporating GO into the selective layer during interfacial polymerization (IP) [11] or by grafting a GO layer onto the existing membrane surface [12]. The problems of the first approach are associated with the necessity of providing uniform filler distribution and avoiding the formation of defects during the IP process in the presence of filler. Thus, the second approach associated with modifying the surface of existing membranes seems more promising. The advantage of surface modification over the formation of mixed-matrix membranes using graphene oxide was demonstrated for PVDF membranes by Wang et al. [13]. However, physically applied coatings can be susceptible to removal during prolonged operation. Therefore, ensuring a strong bond between the GO nanoflakes and the membrane surface, either through charge interactions or covalent cross-linking, is essential. To address this, the simultaneous grafting by polyacrylic acid with the addition of GO dispersion into the monomer solution was carried out for a polyamide RO membrane, resulting in an increase in the flux recovery ratio after bovine serum albumin protein (BSA) fouling from 80% to 98% due to the improved membrane hydrophilicity [14]. Nonetheless, the additional layer formed led to a two-fold decrease in the membrane permeance. Covalent cross-linking using ethylene-diamine (EDA) molecules onto the formed polydopamine layer has demonstrated improved hydrophilicity, surface charge, and resistance against mineral scaling and biofouling [12]. At the same time, the entire polyamide surface can be utilized for GO cross-linking without the formation of an intermediate layer, increasing the membrane resistance. The authors of [15] used the azide-functionalized graphene oxide obtained by the substitution of GO epoxy groups for the modification of a polyamide membrane via a UV-induced cross-linking reaction. This modification altered the surface properties: the water contact angle decreased from 85° to 45°, and the surface roughness decreased from 44.3 to 29.0 nm. As a result, colloidal fouling during prolonged testing was reduced two-fold without significant suppression of the initial membrane permeance after modification. Notably, leveraging the inherent carboxylic groups of polyamide membranes formed due to the hydrolysis of unreacted acyl chloride groups [16] for GO bonding has become a widespread technique [17,18,19,20,21,22,23]. With this method, N-(3-Dimethylaminopropyl)-N’-ethylcarbodiimide hydrochloride (EDC)/N-hydroxysuccinimide (NHS) is employed to convert the native carboxyl groups into intermediate amine-reactive esters, allowing for cross-linking with various diamines followed by cross-linking of the activated GO nanoflakes. This modification improved the membrane hydrophilicity and led to the shielding of the carboxylic groups on the surface of polyamide membranes, which can coordinate multivalent cations with mineral scale formation. This covalent cross-linking enhances the anti-scaling properties and anti-biofouling activity, thanks to GO nanoflakes’ biocidal properties, and provides better fouling resistance toward organic foulants due to the improved membrane hydrophilicity [24].

The commonly described GO deposition procedure involving immersion into a GO suspension needs more precise control over the deposited amount. Few studies have addressed this issue, such as dip-coating polyamide membranes with preliminary activated carboxylic groups into an aminated graphene oxide suspension with different concentrations from 0.05 to 0.3% [25]. This modification improved the antifouling properties of the membrane—the flux decline ratio decreased from 20% to 7.5% due to an increase in the membrane hydrophilicity. Also, layer-by-layer deposition using aminated and pristine graphene oxide nanoflakes was utilized for the modification of RO membranes [26]. The decrease in the water contact angle from 70° to 25° and the surface roughness from 46.5 nm to 21.5 nm after modification reduced the flux decline ratio from 35% to 15% after 12 h of filtration of BSA-containing solution.

At the same time, the amphiphilic nature of GO nanoflakes allows them to anchor at the air–liquid interface, facilitating transfer to solid substrates using the Langmuir–Blodgett or Langmuir–Schaefer techniques [27]. Variation of the compression degree during the deposition allows the formation of diluted, close-packed, or overpacked layers and the transfer of the different amounts of GO with high precision [28]. The density of the graphene oxide layer formed on the air–liquid interface can be easily controlled via monitoring of the surface pressure using the Wilhelmy plate method. This technique has been successfully utilized for the fabrication of transparent conductive coatings [29], luminescent coatings [30] and antibacterial coatings [31]. Thus, this work aims to develop the procedure for modifying commercial reverse osmosis membranes to improve their antifouling properties using the Langmuir–Blodgett approach. The unique feature of this technique is the possibility of forming varied densities of the deposited film, and it can also be easily scaled up and combined with a roll-to-roll process for fabrication of coatings on a large area [32,33]. In the current work, aminated graphene oxide layers assembled at different surface pressure values at the air–liquid interface were transferred to a commercial RO membrane. The microstructure of the transferred layers was characterized using SEM, AFM, and Raman mapping techniques. The antifouling properties of the pristine and modified membranes were studied using a solution containing a model foulant—BSA.

## 2. Experimental

### 2.1. Materials

The commercial Toray (Tokyo, Japan) UTC73HA reverse osmosis membrane was obtained from Sterlitech as a flat sheet. The graphene oxide water dispersion (0.4 wt. %) was procured from Graphenea (Gipuzkoa, Spain). The N-(3-Dimethylaminopropyl)-N’-ethylcarbodiimide hydrochloride (EDC, for synthesis) and ethylenediamine (EDA, >99% purity) were received from Sigma-Aldrich (Sofia, Bulgaria). The anhydrous ethanol was obtained from Merck Co (Darmstadt, Germany). NaCl and the bovine serum albumin (BSA) were procured from Merck Co. for the investigation of the membrane performance.

### 2.2. Membrane Modification

Toray UTC73HA was used for the fouling studies and modification. Aminated graphene oxide was synthesized via the procedure previously utilized in a number of papers devoted to the functionalization of carboxylic groups of carbon-based nanomaterials [34,35,36]. In brief, graphene oxide nanoflakes from a water suspension were deposited using centrifugation at 10,000 rpm, washed with absolute ethanol twice and redispersed in absolute ethanol under ultrasonic treatment. Subsequently, 50 mg of EDC and 500 mg of EDA were added to the 30 mL of graphene oxide ethanol suspension with a concentration of 1 g/L. EDC was added to promote the coupling reaction between the carboxylic graphene oxide groups and the EDA amino group. The scheme of the carboxylic group activation reaction is presented in [37]. In non-aqueous media and in the case of a high primary amine concentration, the activated carboxylic group undergoes fast coupling with the formation of an amide-bond and elimination of the isourea by-product. The mixture was stirred for 15 h at room temperature, followed by three successive centrifugation and washing steps for removing the unreacted EDA. After that, aminated graphene oxide was redispersed in pure water.

For the Langmuir–Blodgett assembly and deposition of aminated graphene oxide, a G2 trough (Kibron) with dimensions of 405 × 80 mm (trough area between the barriers 280 cm^2^) equipped with a dipping well (60 × 60 × 20 mm) was utilized. The trough and barriers were meticulously washed with a soft brush and subsequently rinsed with ethanol and deionized water. The trough was then filled with the water subphase. To prepare the spreading dispersion, 0.375 mL of aminated GO suspension with a concentration of 1 g/L (total amount of spread GO equal to 375 µg) was diluted with 15 mL of a methanol/water solution with a component ratio of 5:1 [27]. The dispersion was spread using a syringe at a rate of 1 mL/min. After spreading, the film at the air–liquid interface was sustained for 20 min before measuring the compression isotherm and transferring it to the substrate. The compression isotherm was collected at a rate of 45 cm^2^/min, and the transfer of the compressed layer to the pre-immersed substrate was carried out with a pull rate of 5 mm/min. For the study of the compressed layer density and microstructure, it was transferred to the SiO_2_/Si wafer piece (typically 10 × 10 mm). To improve the antifouling properties, the aminated GO layer was transferred to the UTC73-HA membrane with preliminary activated carboxylic groups. The carboxylic group activation was carried out using the EDC/NHS technique to suppress the hydrolysis of the intermediate product of the reaction between the carboxylic group and EDC [23]. Before deposition, polyamide membrane coupons of 3 × 3 cm were immersed in a solution containing 4 mM EDC, 10 mM NHS, 10 mM 2-(N-morpholino) ethanesulfonic acid (MES monohydrate), and 0.5 mM NaCl for 60 min. Afterward, the membrane was washed with ultrapure water and immersed in a water subphase for deposition. The transfer of the aminated GO layer from the air–liquid interface to the membrane was performed at surface pressures equal to 10, 20, 30, and 40 mN/m, as monitored by the Wilhelmy plate method, and the obtained samples were labeled as P10, P20, P30, and P40, respectively. After the transfer, the membrane was dried and stored overnight before the fouling tests.

### 2.3. Membrane Characterization

For the characterization of the functional groups in the pristine and amino-modified graphene oxide, the Fourier transform infrared (FTIR) spectra were acquired in the range of 4000 to 500 cm^−1^ using a Nicolet iS5 with an iD7 attenuated total reflectance (ATR) accessory spectrometer (Thermo Scientific, Waltham, MA, USA) by averaging 64 scans captured with a resolution of 4 cm^−1^. The microstructure of the aminated graphene oxide layer transferred to the silicon substrate, as well as the microstructure of the initial and modified RO membranes, was analyzed using an FEI Quanta 450 FEG scanning electron microscope (SEM). Before the SEM characterization, the samples of polyamide membranes were covered by a thin gold layer using magnetron sputtering. The surface roughness was examined using a Brucker Icon Atomic Force Microscope in the QNM (Quantitative Nano Mechanics) mode using a Scanasyst Air tip (Bruker, Billerica, MA, USA). The surface roughness of the membranes was analyzed using Gwyddion software (version 2.66, Brno, Czech Republic).

The uniformity of the graphene oxide layer deposition was studied through Raman mapping experiments using a WITec Alpha 300 Raman microscope (Ulm, Germany) equipped with a 633 nm laser [38]. The Raman maps were recorded over 190 × 190 μm scan areas, measuring one spectrum for 5 microns using a laser spot with a size of 5 µm and a power of 0.2 mW in the True surface mode. Analysis of the spectral data was carried out using WITec Suite SIX software (version 6.1.11.136, Ulm, Germany) to extract the positions and the intensities of the characteristic bands at ~1350 and ~1590 cm^−1^. The water contact angle was measured using the sessile drop technique with the DSA100 setup (Krüss, Hamburg, Germany).

The composition of the modified graphene oxide nanoflakes was studied using X-ray photoelectron spectroscopy on an SPECS (Berlin, Germany) instrument using MgK-α excitation (Eex = 1254 eV). The spectra were calibrated to pure graphite C1s energy (284.6 eV) and processed with Casa XPS software (version 2.3.25PR1).

### 2.4. Filtration Tests

The performance and antifouling properties of the initial and modified polyamide membranes were characterized using a bench-top Sterlitech HP4750 (Auburn, AL, USA) filtration cell with a membrane active area of 3.14 cm^2^. Before the fouling experiments, all the membranes were pretreated by soaking in a 50:50 vol. % water/propanol solution to remove any additives from the membrane surface [39]. To eliminate compaction effects, membrane compaction was performed for 4 h using DI water at a transmembrane pressure of 18 bar. According to our experience and the literature data [40], compaction under hydraulic pressure 1.5 times higher than the operational pressure during 3–4 h will be enough for stabilization of the membrane structure. Further tests of the membrane performance were conducted at a transmembrane pressure of 12 bar. For the continuous monitoring of the membrane permeance, the permeate mass was monitored with a balance WTC 600 (Radwag, Radom, Poland) with a frequency of 1 s. For the determination of the membrane permeance, the mass vs. time dependence was interpolated with 10 min steps, recalculated to the permeate volume using solution density, and then differentiated. The membrane permeance was calculated according to the following equation:(1)$$F=\frac{1}{A\cdot \Delta P}\cdot \frac{dV}{dt}$$
where *A*—membrane area, Δ*P*—transmembrane pressure and $\frac{dV}{dt}$—volume derivative. The membrane rejection toward the salt solution was calculated using the following equation:(2)$$R=1-\frac{C_{p}}{C_{f}}$$
where *C_p_* and *C_f_*—concentration in permeate and feed, respectively, determined from the solution conductivity.

The antifouling membrane characteristics, such as the flux decline ratio and flux recovery ratio, and the contribution of reversible and irreversible fouling were determined in the following manner. The permeance of a 2 g/L NaCl solution was measured for the characterization of the initial membrane permeance (*F_w_*_0_). After filtration of 2 g/L NaCl solution for 60 min, BSA was added to the feed stream to achieve a concentration of 0.2 g/L as a model foulant object. The membrane permeance was measured during 180 min, and the membrane permeance for the solution containing foulant was determined at the end of the experiment (*F_w_*_1_). Then, the membranes were washed with DI water for 5 min, and the membrane permeance (*F_w_*_2_) was measured again using a 2 g/L NaCl solution. Using the obtained data, the flux decline ratio and the flux recovery ratio contributions of reversible and irreversible fouling were calculated using the following equations:(3)$$FRR\left(\%\right)=\left(\frac{F_{w2}}{F_{w0}}\right)\cdot 100$$
(4)$$FDR\left(\%\right)=\left(\frac{F_{w0}-F_{w1}}{F_{w0}}\right)\cdot 100$$
(5)$$DRir\left(\%\right)=\left(1-\frac{F_{w2}}{F_{w0}}\right)\cdot 100$$
(6)$$DRr\left(\%\right)=\left(\frac{F_{w2}-F_{w1}}{F_{w0}}\right)\cdot 100$$

For the evaluation of the long-term stability of the membranes, the fouling–cleaning cycle was performed 3 times for the membrane that demonstrated the best antifouling properties and the pristine membrane.

## 3. Results and Discussion

XPS analysis was utilized to characterize the chemical composition of the pristine and aminated graphene oxide (Figure 1a–c). According to the XPS analysis, the overall C/O ratio for pristine graphene oxide is 1.98. The oxygen atoms are primarily single-bonded to carbon atoms (C–O, corresponding to epoxy and hydroxyl groups). The fraction of this bond with an energy of ~286.5 eV is 54.1%. The contribution of carbon double-bonded to oxygen atoms (C=O, carbonyl groups) with an energy of ~288.6 eV is significantly lower at 4.3%. The content of the C-C bond with an energy of ~284.6 eV is 41.6%. As a result of the amination of graphene oxide, a peak corresponding to the N1s level appears in the overall XPS spectrum, indicating that the overall nitrogen content in the aminated GO is 5.8%. The amination reaction increases the C/O ratio to 2.26 due to the coupling of the ethylenediamine molecule, which leads to a slight increase in the C–C bond content to 44.9%. The contributions of C–O (~286.8 eV) and C–N (~287 eV [41]) cannot be clearly deconvoluted from the C1s region due to their close energies. However, an additional peak appears at ~289.1 eV in the C1s region, corresponding to the O=C–N bond, with a contribution of 2.7%. This value is in good agreement with the content of carboxylic groups in graphene oxide, which is 3.2% [42]. Thus, we can conclude that the successful binding of ethylenediamine molecules with carboxylic groups has taken place, achieving a reaction yield of more than 80%.

FTIR spectroscopy (Figure 1d) also confirmed the modification of graphene oxide nanoflakes with diamine molecules. Pristine GO exhibits characteristic peaks at 1050 cm^−1^ (CH–OH stretching), 1230 cm^−1^ (Symmetric C–O–C ring breathing), 1380 cm^−1^ (–OH bending), 1620 cm^−1^ (H–O–H bending), 1720 cm^−1^ (C=O symmetric stretching) and a wide band at ~3200 cm^−1^ (O–H stretching) [43]. These characteristic peaks indicate the presence of hydroxyl, epoxy, and carboxylic functional groups. Amination of graphene oxide significantly reduces the peak intensity at 1720 cm^−1^. The residual intensity of this peak corresponds to the presence of unreacted carboxylic groups with a relative amount of less than 20%, which was determined by XPS analysis. Additionally, a new peak at 1160 cm^−1^ indicates the formation of O–NH bonds. A slight intensification of the board band at ~3200 cm^−1^ and a shift in the peak position to higher wavenumbers suggest the presence of N-H vibrations. Therefore, the FTIR results validate the modification of graphene oxide with ethylenediamine molecules. These aminated graphene oxide nanoflakes can be utilized for covalent cross-linking to the surface of activated polyamide membranes.

The Langmuir–Blodgett technique was chosen to modify the reverse osmosis membranes, as it allows precise tuning of the nanoflake density transferred to the substrate. The density of the nanoflakes at the air–liquid interface can be tuned by moving the barriers and monitoring the surface pressure with a tensiometer. The measured surface pressure–trough area isotherms for the pristine and aminated graphene oxide are presented in Figure 2. For a trough area higher than 45%, moving the barrier does not lead to surface pressure variation, indicating the presence of diluted, well-isolated individual GO nanoflakes at the air–liquid interface. A sharp increase in surface pressure occurs when the compression degree reaches 45% and 40% of the trough area for the aminated and pristine graphene oxide, respectively. For the characterization of the microstructure of the GO layers, they were collected at different degrees of compression by vertically drawing the silicon substrate pre-immersed into the subphase. The microstructure of the obtained coating was characterized by SEM and is presented in Figure 2. One can observe that at low surface pressure (Π = 10 mN/m), the structure with large voids between the aGO nanoflakes was formed. A further increase in the surface pressure to 20 mN/m leads to the compaction of the aGO layer and a decrease in the area of voids. Compression to the surface pressure of 30 mN/m leads to the assembly of a dense layer with nanoflakes overlapping and the presence of many wrinkles. Further compression leads to the growth of the surface pressure, in contrast to a monolayer of surfactants, which, collapsing to a multilayer, results in a surface pressure drop. As a result of higher surface pressure (Π = 40 mN/m), a large number of wrinkles and folded nanoflakes are formed in the transferred layer, increasing its roughness. The difference in the microstructure of the compressed layers of pristine and aminated graphene oxide nanoflakes is more pronounced at high surface pressures of a compressed layer. SEM micrographs are presented as insets in the surface pressure–trough area isotherms. It should be noted that in contrast with the initial GO, we observe a large number of overlapping nanoflakes and wrinkles for the aminated GO. This can be explained by the decrease in the aggregation stability of GO after amination due to the reduction of the charge after the conversion of carboxylic groups at the edges of nanoflakes into amino groups. The presence of charged carboxylic groups leads to a strong in-plane repulsion for pristine graphene oxide [44]. As a result, the flakes’ surface concentration for pristine GO is lower than for aminated GO, and the surface pressure is lower for an equal compression degree (left shift of the compression isotherm for pristine GO). Moreover, the strong in-plane repulsion results in the presence of voids in the transferred GO layer, even at a high surface pressure of 40 mN/m.

Thus, the variation of the surface pressure during Langmuir–Blodgett deposition can be effectively utilized to vary the density of the graphene oxide layer transferred to the membrane surface. On the one hand, the increase in the density of the deposited GO should improve the membrane hydrophilicity, leading to improved membrane antifouling properties. On the other hand, the deposition of a dense GO layer may result in the growth of membrane resistance and the suppression of its permeance. Thus, for further study, the aminated GO layers assembled at the air–liquid interface at varied surface pressures of 10, 20, 30, and 40 mN/m were transferred to the surface of the UTC73-HA membrane with pre-activated carboxylic groups.

The initial and modified UTC73-HA membranes’ microstructure was first characterized using SEM (Figure 3). The selective layer of the pristine membrane demonstrates leaf-like morphology, which is common for RO membranes. Transfer of the graphene oxide layer to the membrane surface allows the smoothing of the polyamide layer. The island-like structure of the graphene oxide flakes is formed on the membrane surface at low surface pressures (Π = 10–20 mN/m). At the same time, almost the whole surface of the RO membrane is covered with GO nanoflakes at high surface pressures (Π = 30–40 mN/m). Also, graphene oxide corrugations and folding are present on the coating formed at Π = 40 mN/m.

To characterize the roughness of the formed coating, which may have an influence on the membrane’s antifouling properties, atomic force microscopy was utilized [45]. The AFM image (Figure 3) of the pristine UTC73-HA membrane exhibits the familiar ridge-and-valley structure of polyamide, possessing high surface roughness (values of the surface roughness are presented in Figure 4a). Transferring the aGO layer assembled at low surface pressure to a polyamide surface led to reduced roughness, while the growth of the surface pressure led to the growth of the formed coating roughness due to the folding and overlapping of GO nanoflakes and the formation of wrinkles. The values of the root mean square roughness (S_q_) and arithmetical mean height (S_r_) calculated from the AFM images of the aminated graphene oxide coating assembled at different surface pressures and transferred to the membrane surface are presented in Figure 4a.

It should be noted that a quantitative comparison of the thickness and distribution of the GO layer along the membrane surface cannot be precisely performed using SEM or AFM techniques due to the high locality of these methods. Thus, further Raman spectroscopy mapping was utilized for the estimation of the uniformity of the GO coatings. The distribution of the G-mode integral intensity (Raman shifts ranges 1450–1690 cm^−1^) along a 380 × 380 μm area is presented in Figure 3. Based on the obtained data, the average integral intensity and standard deviation of the Raman intensity were calculated. The dependence of these parameters on the surface pressure of the assembled coating is shown in Figure 4a. The results of the Raman mapping support the SEM observation that the GO coating demonstrates an island-like structure for low values of surface pressures, associated with the very high standard deviation of the integral intensity. The increase in surface pressure results in the growth of the signal integral intensity from 33.6·10^5^ to 67.8·10^5^ counts for the GO layer transferred at 10 mN/m and 40 mN/m, respectively, and the decrease in the relative standard deviation. The coating transferred at a surface pressure equal to 40 mN/m demonstrates high uniformity.

The hydrophilic nature of GO should lead to the improvement of the modified membrane’s hydrophilicity; indeed, the water contact angle (WCA) for the pristine membrane decreases from 67° to 55° after modification of the membrane surface with the dense GO layer assembled at 40 mN/m. The dependence of the WCA on the surface pressure of the GO layer assembly is shown in Figure 4b. The increased membrane hydrophilicity suggests the improvement of the antifouling properties toward organic fouling [46].

The desalination performance of the commercial and modified membranes was tested using a 2 g/L NaCl solution (Figure 5a). The pristine membrane demonstrated the highest water permeance of 2.70 m^3^/(m^2^ bar h); its covering with the graphene oxide layer resulted in permeance suppression due to additional layer resistance. At the same time, the formed GO layer slightly increased the NaCl rejection from 95.2% to 98.1% due to the covering of defects in the polyamide layer. Salt solution containing 0.2 g/l BSA was utilized for the study of the antifouling properties of the obtained membranes. From Figure 5b, one can observe that the water permeability of all the membranes reduced for the BSA/salt solution compared to the pure salt solution. This effect is governed by the deposition of BSA molecules on the membrane surface. To quantitatively describe this effect, we used the flux decline ratio values—the pristine membrane lost 38% of the initial permeance during the filtration of the BSA-containing solution (Figure 5c). The membrane covered with the GO layer caused the suppression of the deposition of BSA molecules and decreased the FDR down to 18% for the aGO layer transferred at 20 mN/m surface pressure, while the increase in the aGO layer density resulted in the growth of the FDR values to ~30%.

The effect of BSA adsorption was partially reversible—adsorbed molecules can be removed using membrane washing. This procedure partly recovers the membrane permeability—after washing, the water flux increases. The relation between the membrane permeability after washing and the pristine membrane permeability, called the flux recovery ratio, can be used for the characterization of the degree of irreversible fouling. For the unmodified polyamide membrane, the flux recovery ratio is equal to 65.1%. Coating the polyamide membrane with an aGO layer reduced the irreversible fouling—the FRR increased up to 77.2% for the P10 membrane and to 87.2% for the P20 membrane, which can be explained by both the growth of the hydrophilicity and the smoothing of the membrane surface. Further increasing the density of the transferred aGO layer leads to reducing the FRR to 76.8% and 74.2% and increasing the flux decline ratios to 30.6% and 32.5% for the P30 and P40 membranes, respectively. This effect can be explained by the growth of the surface roughness of the formed coating in comparison with the P20 membrane. Typically, an increase in the membrane surface roughness results in the growth of the foulant deposition rate [45]. Also, the improved surface hydrophilicity of the membranes coated with graphene oxide results in the growth of the reversible fouling contribution compared to the pristine polyamide membranes. In spite of the improved antifouling properties of the P30 and P40 membranes, their permeance after the fouling test appears to be lower than the permeance of the pristine RO membrane. This is caused by the excess resistance of dense aGO layer; thus, from the practical point of view, these membranes are not of interest. At the same time, after the fouling test, the permeance of the P10 and P20 membranes is higher than the pristine membrane—1.88, 1.92 and 1.69 L/(m^2^ bar h), respectively. Based on the performance and antifouling test results, we can conclude that the RO membrane modified by transferring an aminated GO layer formed by compression at 20 mN/m surface pressure demonstrates the best antifouling performance and higher permeance values during long-term antifouling tests.

The long-term stability of the formed coatings was studied during three fouling–cleaning cycles, each 180/60 min (Figure 6a). For the modified membrane P20, after the second cycle, the normalized permeance stabilized at a constant level, while for the pristine membrane, we observed a progressive drop in permeance. These results indicate the improvement of the antifouling properties. The stability of the obtained surface coating and the absence of leaching out of the cross-linked graphene oxide nanoflakes was proved using the Raman mapping technique after a prolonged filtration test, including fouling and cleaning steps (Figure 6b). The average intensity of the G-mode Raman signal calculated from the obtained map is equal to (38.7 ± 15.9) × 10^3^ counts, which is close to this value for the initial P20 membrane given. Also, a membrane autopsy study after filtration was performed using SEM (Figure 6c). According to the SEM analysis, the graphene oxide layer remains on the membrane surface.

We compared our results with those previously published in the literature, which examined the enhancement of antifouling characteristics against BSA foulant concentrations ranging from 100 to 1000 mg/L through the modification of RO membranes. From Table 1, through comparison of the properties of pristine RO membranes and membranes after modification, it can be observed that surface modification strategies typically result in a reduction in permeance by 14–50%, while an increase in selectivity is noted. Additionally, the hydrophilic nature of the modifiers used significantly reduces both irreversible and reversible fouling. This is evidenced by an increase in the flux recovery ratio (FRR) and a decrease in the flux decline ratio (FDR) compared to the pristine membranes. Our results are comparable with those previously published; surface modification led to a 17% decrease in the membrane permeance, an increase in the NaCl rejection, and an improvement in the FRR from 66% to 86%. Also, in Table 1, we analyze the results achieved using the approach associated with the creation of TFN membranes. Here, pristine membranes refer to polymeric membranes formed without the addition of modifiers during the interfacial polymerization step. The introduction of nanoparticles into the polymer matrix improves both the permeability and antifouling properties. However, this way requires the development of new synthetic approaches and careful control of the membrane synthesis process to avoid the formation of defects that suppress the rejection characteristics.

In the current work, we have proposed a technique for the formation of stable graphene oxide coatings on the surface of commercial reverse osmosis membranes, allowing an increase in the membrane antifouling characteristics. According to our experimental results, the aminated GO coating formed at a surface pressure equal to 20 mN/m and transferred to the commercial reverse osmosis membrane UTC73-HA by the Langmuir–Blodgett technique allows for improved membrane antifouling properties during the filtration of a BSA-containing solution—the flux decline ratio decreases from 38% to 18%, and the flux recovery ratio increases from 65% to 87%. This modification does not significantly suppress the membrane permeance, which decreases from 2.7 to 2.28 L/(m^2^ bar h) after modification. The performed autopsy study of the membrane after the filtration experiment demonstrates the stability of the formed GO coating without significant leaching of graphene oxide. It should be noted that the Langmuir–Blodgett process can be easily scaled up and merged with a roll-to-roll process for coating fabrication on large areas [32,33], and the low modifier consumption is an undoubted advantage of this process. The modifier consumption can be estimated based on the experimental data; for the coating formation, 375 μg of aminated graphene oxide was spread on a subphase area of 280 cm^2^. The best antifouling properties were demonstrated for the coating formed at a surface pressure of 20 mN/m, corresponding to a trough area of 70 cm^2^ (relative trough area 25%). Thus, we estimate that forming an antifouling coating on 1 m^2^ requires about 53 mg of graphene oxide. Moreover, the modifier consumption can be reduced by further optimization of the surface properties of graphene oxide [54].

## 4. Conclusions

The commercial reverse osmosis membrane UTC73-HA was coated with a layer of aminated graphene oxide using the Langmuir–Blodgett technique at different degrees of compression of the film on the air–liquid interface. This allowed for deposition of a different amount of aGO and variation of the coating morphology from the island-like structure to the uniform coating. The stability of the formed coating was ensured by the covalent cross-linking of the aminated graphene oxide nanoflakes and hydroxyl groups on the membrane surface, pre-activated with EDC/NHS. This modification allowed for improving the membrane hydrophilicity and reducing the surface roughness, resulting in the increase in the membrane fouling stability toward colloidal fouling. The flux recovery ratio after modification increased from 65% to 87%, retaining high permeability. However, a significant increase in the coating density suppressed the membrane permeance, making membranes modified at high surface pressures uninteresting from a practical point of view. At the same time, the membrane modified by the aminated GO layer transferred at a surface pressure of 20 mN/m demonstrated higher performance during long-term fouling tests in comparison with the pristine membrane. Also, the low modifier consumption is an advantage of the suggested technique, which allows for better control of the adsorbed GO mass compared with simply dip coating, and it can be easily scaled using the roll-to-roll approach for coating large-area membranes.

## Figures and Tables

**Figure 1 membranes-14-00172-f001:**
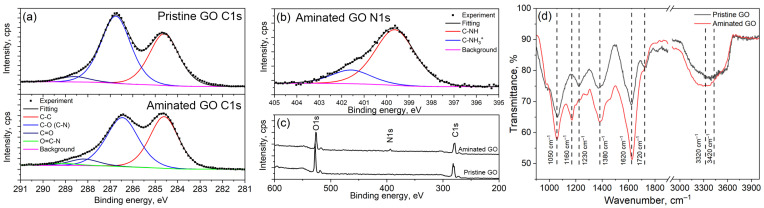
Results of the XPS analysis of pristine and aminated graphene oxide: C1s (**a**), N1s (**b**) and overall (**c**) spectra. (**d**) ATR-FTIR spectra of pristine and aminated graphene oxide.

**Figure 2 membranes-14-00172-f002:**
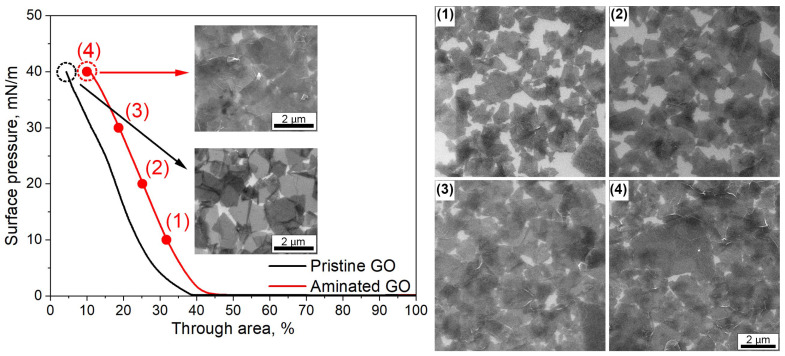
Typical compression isotherms of pristine and aminated graphene oxide spread on the air–liquid interface. The insets demonstrate the microstructure of the transferred layer at the surface pressure of 40 mN/m. SEM images of aminated graphene oxide transferred on the silicon substrate at surface pressures (Π) equal to 10 (**1**), 20 (**2**), 30 (**3**) and 40 (**4**) mN/m.

**Figure 3 membranes-14-00172-f003:**
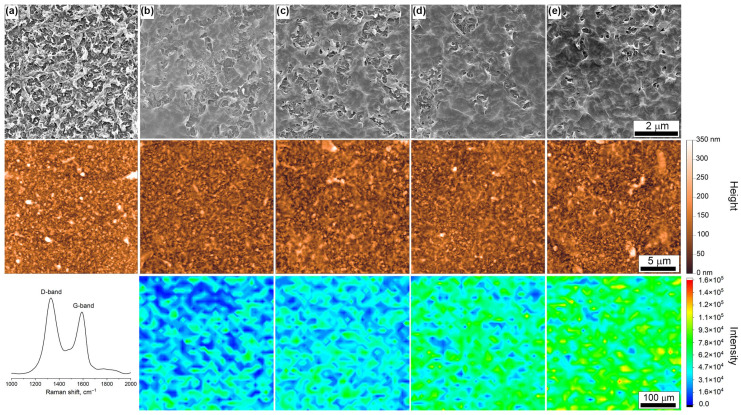
Microstructure of pristine UTC73-HA (**a**) membrane and membranes covered with aminated graphene oxide layer transferred using Langmuir–Blodgett technique from the air–liquid interface at surface pressures of 10 (**b**), 20 (**c**), 30 (**d**), and 40 (**e**) mN/m characterized using SEM, AFM, and Raman mapping of G-mode intensity.

**Figure 4 membranes-14-00172-f004:**
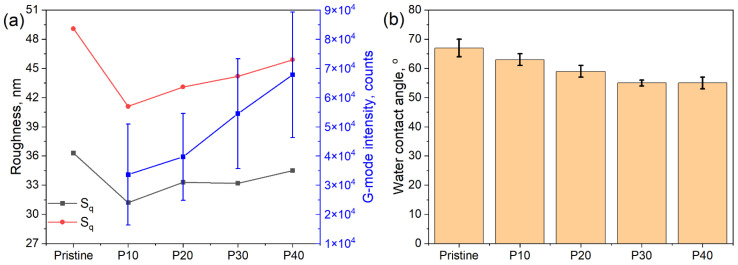
Surface roughness calculated from the AFM scans and integral intensity of the G-mode Raman signal (**a**), and water contact angle (**b**) measured for pristine RO membranes and RO membranes modified with aGO layer assembled at different surface pressures.

**Figure 5 membranes-14-00172-f005:**
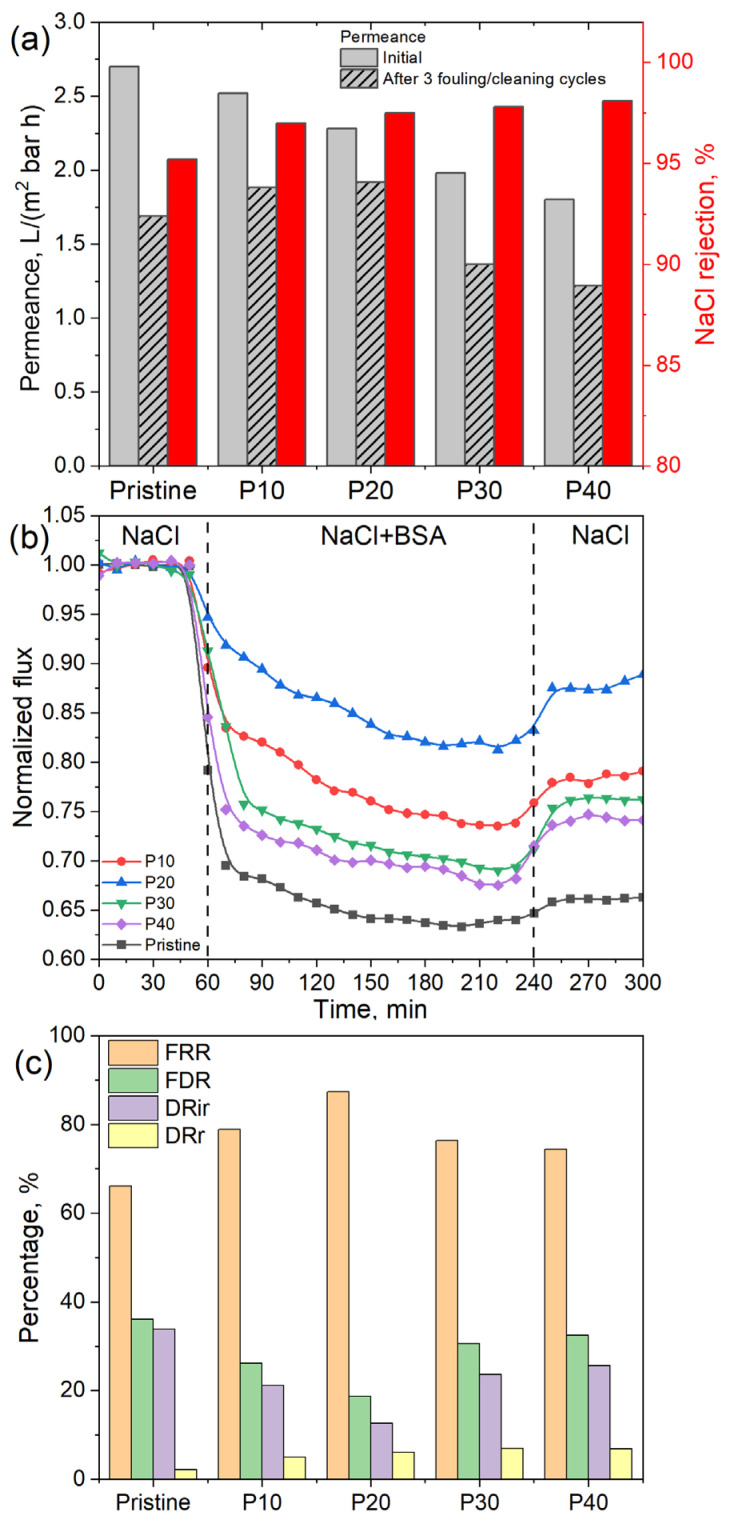
Results of the permeance and salt rejection study (**a**) and the study of the membrane performance (in terms of the permeance normalized to the initial permeance) under fouling conditions. These conditions were assessed by sequentially measuring the permeance of a 2 g/L NaCl solution and a solution containing 2 g/L NaCl and 0.2 g/L BSA, with intermediate washing of the membrane with DI water for 5 min (**b**). Comparison of the fouling characteristics of the obtained membranes: flux recovery ratio (FRR), flux decline ratio (FDR), reversible (DRr), and irreversible (DRir) flux decline ratios (**c**).

**Figure 6 membranes-14-00172-f006:**
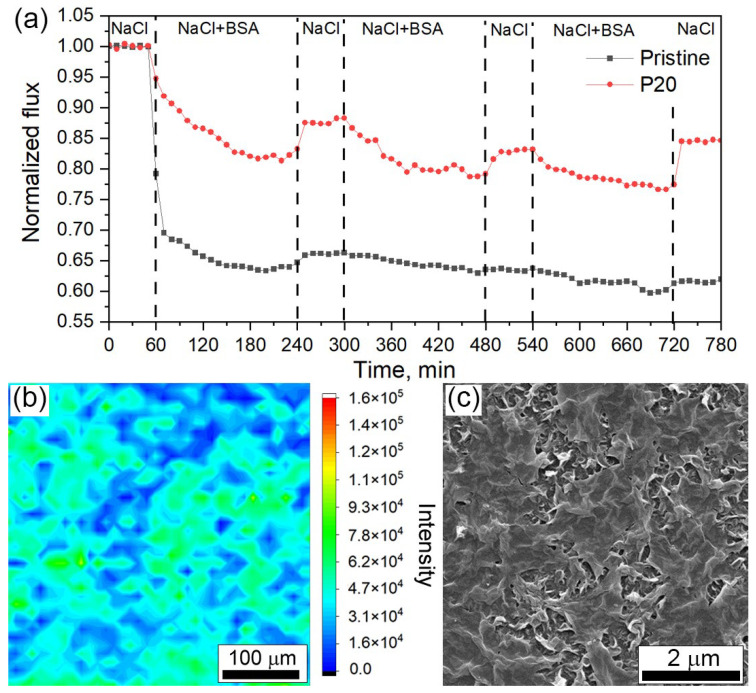
Results of the long-term performance study for the pristine and modified membrane during 3 fouling–cleaning cycles (**a**). Microstructure of the P20 membrane after fouling study characterized using Raman mapping of the G-mode intensity (**b**) and SEM (**c**).

**Table 1 membranes-14-00172-t001:** Published data concerning the improvement of the antifouling properties of RO membranes via surface modification or formation of thin composite membranes.

Modifier	Permeance, L/(m^2^ bar h)		NaCl Rejection, %		FDR, % (Time, h)		FRR, %		Ref.
	Pristine	Modified	Pristine	Modified	Pristine	Modified	Pristine	Modified	
Surface modification									
GO	2.3	1.1	95%	98%	25.5% (3)	10% (3)	80%	98%	[14]
AgNP	5	4.6	98%	97%	3% (5)	9% (5)	93%	98%	[47]
Dopamine/tannic acid	4.9	3.6	97.2%	98.5%	23% (5)	9% (5)	82%	96&	[48]
Ferric-phytic acid	4.5	3.5	96.2%	98%	35%(24)	12%(24)	75%	94%	[49]
TiO_2_ nanosheets	1.06	1.39	97%	97.2%	36% (5)	20% (5)	87%	100%	[50]
GO	2.6	2.25	94.1%	95.3%	65% (168)	43% (168)	n/a	n/a	[15]
GO	2.7	2.28	95.2%	98%	38% (3)	18% (3)	65%	87%	This work
Thin film composite membranes									
CQD/Ag	1.69	2.44	98.6%	98.9%	44% (20)	36% (20)	68%	83%	[51]
CNT	0.62	1.45	98.2%	98.1%	38% (10)	24% (10)	72%	100%	[52]
GO	0.2	0.42	97.6%	81%	70% (12)	45% (12)	92%	98%	[53]

GO—graphene oxide, CNT—carbon nanotube, CQD—carbon quantum dot, AgNPs—silver nanoparticles.

## Data Availability

The raw data supporting the conclusions of this article will be made available by the authors on request.

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
