# Peer review of "Graphene Oxide Surface Modification of Reverse Osmosis (RO) Membrane via Langmuir–Blodgett Technique: Balancing Performance and Antifouling Properties"

_membranes, 2024, doi:10.3390/membranes14080172_

Round 1
Reviewer 1 Report
Comments and Suggestions for Authors
This is a well-organized and written manuscript. I am glad to see the authors used a commercial membrane to modify their surface using GO via the Langmuir-Blodgett technique. The following are comments/questions the authors should address before accepting this manuscript for publication.
1) The base commercial membrane is LPRO. Using 12 bar (18 bar for precompaction) seems low. Why this pressure was selected? Is there any stability issue with GO NPs at higher pressures? Also, higher pressures would increase the apparent rejection.
2) I like Table 1, but the authors should also provide information about the pristine membrane in each study
3) TFN membranes' improved fouling resistance is evident. However, because the pristine membrane has a higher initial permeance, its final permeance is still comparable to that of the best TFN membrane (Fig. 5a). Are the reported differences in Fig. 5a statistically significant?
4) What about the effect of fouling on salt rejection and its recovery?
Author Response
Reply to Reviewer #1
Thank you for the close attention to our manuscript. We glad to hear your opinion that our manuscript is well-organized and written. Please find our reply to your questions below.
- Reviewer: The base commercial membrane is LPRO. Using 12 bar (18 bar for precompaction) seems low. Why this pressure was selected? Is there any stability issue with GO NPs at higher pressures? Also, higher pressures would increase the apparent rejection.
Authors: According to the specification given by manufacture (https://www.sterlitech.com/toray-flat-sheet-membrane-utc-73ha-pa-ro-47mm-5-pk.html) the main treatment object for Toray UTC-73HA membrane is brackish water. Typically, the operation pressure for brackish water reverse osmosis is 12 bar (and about 6-7 bar for low pressure reverse osmosis) [1]. Thus, we choose the operation pressure equal to 12 bar for treatment the 2 g/L NaCl solution, which simulates the brackish water. According to our experience and literature data [2], the compaction under the hydraulic pressure 1.5 times higher than operational pressure during 3-4 h will be enough for stabilization membrane structure.
Multilayer graphene oxide membranes can be compressed by the external pressure in several conditions (such operation in humid air) due to the variable interlayer distance [3]. However, in our case Langmuir-Blodget technique allows to transfer single-layer nanosheets to the membrane surface which is stable towards pressure compaction and chemical cross-linking with polyimide layer provide their stability against leaching off during filtration.
Yes, you are right – increase of operation pressure in reverse osmosis process typically increase the rejections characteristics of membranes, but the increase of the operation pressure results in growth of energy consumption of overall process.
Action taken: A few clarifying remarks were added to the text.
- Reviewer: I like Table 1, but the authors should also provide information about the pristine membrane in each study
Authors: Data listed in the table 1 as data for pristine membranes in the case of surface modification strategy corresponds to the characteristics of initial polymeric membranes. While in the case of thin film composite membranes pristine membranes are understood as polymeric membranes formed by the same procedure, but without modifier addition during interfacial polymerization step.
Action taken:. The description of table 1 was slightly modified.
- Reviewer: TFN membranes' improved fouling resistance is evident. However, because the pristine membrane has a higher initial permeance, its final permeance is still comparable to that of the best TFN membrane (Fig. 5a). Are the reported differences in Fig. 5a statistically significant?
Authors: Membrane permeance after 3 fouling/cleaning cycles is equal to 1.68 and 1.92 l/(m2 bar h) for pristine and P20 modified membranes, respectively. In our opinion the 15% difference is a measurable value, illustrating the effect of surface modification. Moreover, we added the long-term fouling test data, illustrating modified membrane preserve their characteristics throughout 3 fouling cleaning cycles, while for unmodified membrane we observe a progressive drop of membrane permeance (see also the reply to question 10 of reviewer 2).
Action taken:. We provide the additional data for 3 fouling/cleaning cycles (Figure 6a) to compare the long-term performance of pristine and modified membranes.
- Reviewer: 4) What about the effect of fouling on salt rejection and its recovery?
Authors: We did not observe a significant difference in salt rejection between the pristine and fouled membranes. The difference was less than 0.3%, which can be attributed to random error.
Reviewer 2 Report
Comments and Suggestions for Authors
This study utilizes the Langmuir-Blodgett technique to modify the surface of reverse osmosis membranes in order to enhance their antifouling properties. The changes in properties and morphology of the membranes modified at various surface pressures were investigated. However, some experimental phenomena are not fully explained. The specific experimental process was not described in detail, and the antifouling properties of the membranes were not sufficiently highlighted. The data presented in the figure are not clear enough to distinguish between the different membranes. Therefore, a major revision is recommended before publication. Detailed comments are as follows:
1. (formula. 2) The calculation formula is incorrect.
2. (Fig. 1 and Fig. 4a) The corresponding data in the figure should be clearly labeled, and a clear distinction should be made between pristine graphene oxide and aminated graphene oxide.
3. (Page 6, lines 246-248) “Amination of graphene oxide results in the disappearance of the peak at 1720 cm−1, confirming the formation of amide bonds between carboxylic groups and ethylenediamine”. The peak at 1720 cm-1 has not disappeared. A more specific and detailed description of this point would be appreciated.
4. (Page 6, lines 263-264) “A gradual increase in surface pressure occurs when the compression degree reaches 45% of the through area.”. This statement does not correspond to the data in Figure 2. Please revise it.
5. (Page 6, lines 269-271) “It should be noted that for aminated GO, in contrast with initial graphene oxide flakes [42], overlapping of nanoflakes occurs.”. Please provide SEM images demonstrating the differences in the initial graphene oxide flakes.
6. (Page 7, line 284-285) “On the other hand, the deposition of a dense GO layer may result in the growth of membrane resistance and the suppression of its permeance.” Please provide a chart illustrating the increase in membrane resistance.
7. (Page 8, lines 309-310) “The AFM image (Figure 3) of the pristine UTC73-HA membrane exhibits the familiar ridge-and-valley structure of polyamide, possessing high surface roughness.”. Please provide the roughness values from Figure 3 to visualize the changes in roughness on the membrane surface.
8. (Page 9, line 354-357) “and the low modifier consumption is an undoubted advantage of this process.” What is the basis for low modifier consumption?
9. (Page 12, line 432-433) “The membrane covering with the GO layer caused suppression of the deposition of BSA molecules and decreased FDR down to 18% for the aGO layer transferred at 20 mN/m surface pressure, while the increase of the aGO layer density resulted in the growth of FDR values to ~30%.” The description of FDR growth up to 30% is unclear. The explanation between the two needs to be more specific.
10. Please supplement the long-term performance data from the antifouling test with data from at least two cycles to assess the stability of the membrane over time.
11. Please use colors with higher contrast for labeling in SEM images.
12. These two papers may assist the authors in exploring the impact of anti-contamination strategies on membrane performance. (Advanced Membranes 4 (2024): 100095; Advanced Membranes 2 (2022): 100038)
Author Response
Reply to Reviewer #2
Dear Reviewer! Thank you for the close attention to our manuscript. Your comments and recommendations allow us to improve the manuscript quality and to highlight the effects of surface modification. Please find our reply to your questions below.
- Reviewer: (formula. 2) The calculation formula is incorrect.
Authors: Thank you for this note, the misprint was corrected.
Action taken: Corrected.
- Reviewer: (Fig. 1 and Fig. 4a) The corresponding data in the figure should be clearly labeled, and a clear distinction should be made between pristine graphene oxide and aminated graphene oxide.
Action taken: Corrected, the required figures was clearly labeled.
- Reviewer: (Page 6, lines 246-248) “Amination of graphene oxide results in the disappearance of the peak at 1720 cm−1, confirming the formation of amide bonds between carboxylic groups and ethylenediamine”.The peak at 1720 cm-1has not disappeared. A more specific and detailed description of this point would be appreciated.
Authors: Thank you for this clarification! We agree that the peak at 1720 cm-1 did not disappear completely, but its intensity significantly reduced relative to other peaks. According to the provided XPS analysis – the reaction yield of polycondensation between carboxylic groups and diamine molecules is about 80%. Thus, we can attribute the residual intensity of this peak to the presence of non-reactive carboxylic groups with an amount less than 20%
Action taken: The more detailed description of 1720cm-1 peak behavior was added to the manuscript.
- Reviewer: (Page 6, lines 263-264) “A gradual increase in surface pressure occurs when the compression degree reaches 45% of the through area.”.This statement does not correspond to the data in Figure 2. Please revise it.
Action taken: Corrected
- Reviewer: (Page 6, lines 269-271) “It should be noted that for aminated GO, in contrast with initial graphene oxide flakes [42], overlapping of nanoflakes occurs.”.Please provide SEM images demonstrating the differences in the initial graphene oxide flakes.
Authors: We added the compression isotherm for pristine graphene oxide in the revised version of the manuscript. However, we do not want to overload the text with additional data about the microstructure of pristine graphene oxide. In our opinion, the difference in microstructure is more pronounced for the coatings formed at a high compression degree—40 mN/m surface pressure. Thus, two insets demonstrate the microstructure of formed layers of pristine and aminated graphene oxide in the compression isotherm graph.
Action taken: We added compression isotherm for pristine GO, micrographs, and additional descriptions in the revised text.
- Reviewer: (Page 7, line 284-285) “On the other hand, the deposition of a dense GO layer may result in the growth of membrane resistance and the suppression of its permeance.”Please provide a chart illustrating the increase in membrane resistance.
Authors: In this sentence, we meant that the deposition of an additional layer of graphene oxide on the surface of the reverse osmosis membrane would reduce the system's overall permeance, which was proved later in the text. A chart demonstrating the contributions of the pristine membrane and formed graphene oxide layer to overall membrane resistance will complicate the manuscript text.
- Reviewer: (Page 8, lines 309-310) “The AFM image (Figure 3) of the pristine UTC73-HA membrane exhibits the familiar ridge-and-valley structure of polyamide, possessing high surface roughness.”. Please provide the roughness values from Figure 3 to visualize the changes in roughness on the membrane surface.
Authors: Roughness values for AFM images are represented on the figure 4a, the clarifying link to the figure 4a is added to the above-mentioned sentence.
- Reviewer: (Page 9, line 354-357) “and the low modifier consumption is an undoubted advantage of this process.” What is the basis for low modifier consumption?
Authors: In our experiments, the spread of aminated graphene oxide amount was equal to 375 mg in the subphase area of 280 cm2. The best antifouling properties was demonstrated for the coating formed at a surface pressure of 20 mN/m, corresponding to the trough area of 70 cm2 (relative trough area 25%). Thus, creating an antifouling coating on 1 m2 requires about 53 mg of graphene oxide; in our opinion, this is relatively low consumption. Moreover, the modifier consumption can be reduced by further optimization of the surface properties of graphene oxide.
Action taken: Clarifying remarks were added to the manuscript text.
- Reviewer: (Page 12, line 432-433) “The membrane covering with the GO layer caused suppression of the deposition of BSA molecules and decreased FDR down to 18% for the aGO layer transferred at 20 mN/m surface pressure, while the increase of the aGO layer density resulted in the growth of FDR values to ~30%.” The description of FDR growth up to 30% is unclear. The explanation between the two needs to be more specific.
Authors: In our opinion, this effect can be explained by the growth of surface roughness with a further increase of the surface pressure higher than 20 mN/m. Typically, an increase in surface roughness results in the growth of foulant deposition rate [4]. This explanation was mentioned further.
Action taken: Clarifying remark with reference was added to the manuscript text.
- Reviewer: Please supplement the long-term performance data from the antifouling test with data from at least two cycles to assess the stability of the membrane over time.
Action taken: Thank you for this suggestion! The graph demonstrating the stability of pristine and modified membranes during three fouling/cleaning cycles was added as Figure 6a. One can observe that continuous permeance dropping occurs for pristine membranes, while for modified membranes, the permeance stabilizes after the second cycle.
- Reviewer: Please use colors with higher contrast for labeling in SEM images.
Action taken: Corrected
- Reviewer: These two papers may assist the authors in exploring the impact of anti-contamination strategies on membrane performance. (Advanced Membranes 4 (2024): 100095; Advanced Membranes 2 (2022): 100038).
Authors: Thank you for these references, they were cited in the revised manuscript.
References
[1] Y.J. Lim, Y. Ma, J.W. Chew, R. Wang, Assessing the potential of highly permeable reverse osmosis membranes for desalination: Specific energy and footprint analysis, Desalination. 533 (2022) 115771. https://doi.org/https://doi.org/10.1016/j.desal.2022.115771.
[2] C. Xu, Z. Wang, Y. Hu, Y. Chen, Thin-Film Composite Membrane Compaction: Exploring the Interplay among Support Compressive Modulus, Structural Characteristics, and Overall Transport Efficiency, Environ. Sci. Technol. 58 (2024) 8587–8596. https://doi.org/10.1021/acs.est.4c01639.
[3] E.A. Chernova, K.E. Gurianov, D.I. Petukhov, A.P. Chumakov, R.G. Valeev, V.A. Brotsman, A. V Garshev, A.A. Eliseev, Oxidized Carbon-Based Spacers for Pressure-Resistant Graphene Oxide Membranes, Membranes (Basel). 12 (2022). https://doi.org/10.3390/membranes12100934.
[4] C. Shang, D. Pranantyo, S. Zhang, Understanding the Roughness–Fouling Relationship in Reverse Osmosis: Mechanism and Implications, Environ. Sci. Technol. 54 (2020) 5288–5296. https://doi.org/10.1021/acs.est.0c00535.
Round 2
Reviewer 2 Report
Comments and Suggestions for Authors
My questions have been answered in the revised version and the current version is recommended to be accepted.